# Unlocking the Potential of Spheroids in Personalized Medicine: A Systematic Review of Seeding Methodologies

**DOI:** 10.3390/ijms26136478

**Published:** 2025-07-04

**Authors:** Karolina M. Lonkwic, Radosław Zajdel, Krzysztof Kaczka

**Affiliations:** 1Clinic of General and Oncological Surgery, Medical University of Lodz, 92-213 Lodz, Poland; krzysztof.kaczka@umed.lodz.pl; 2Mabion S.A., 95-050 Konstantynow Lodzki, Poland; 3Department of Economic and Medical Informatics, University of Lodz, 90-214 Lodz, Poland; radoslaw.zajdel@uni.lodz.pl; 4Department of Medical Informatics and Statistics, Medical University of Lodz, 90-645 Lodz, Poland

**Keywords:** spheroids, personalized medicine, drug screening, organoids, 3D models

## Abstract

Three-dimensional (3D) spheroid models have revolutionized in vitro cancer research by offering more physiologically relevant alternatives to traditional two-dimensional (2D) cultures. A systematic search identifies English-language studies on patient-derived cancer spheroids for drug screening, using defined inclusion and exclusion criteria, with data extracted on cancer type, culture methods, spheroid characteristics, and therapeutic responses. This manuscript evaluates the methods for spheroid formation and the cellular sources used, highlighting the diverse applications and preferences in this field. The five most investigated cancer origins for spheroid seeding are breast, colon, lung, ovary, and brain cancers, reflecting their clinical importance and research focus. Among seeding methodologies, forced-floating and scaffold-based methods predominate, demonstrating reliability and versatility in spheroid generation. Other techniques, including microfluidics, bioprinting, hanging drop, and suspension culture also play significant roles, each with distinct advantages and limitations. This review underscores the increasing use of spheroid models and the need for standardization in methodologies to enhance the reproducibility and translational potential in cancer research.

## 1. Introduction

In recent years, there has been a paradigm shift in biomedical research towards developing more clinically relevant models to study human physiology and diseases. Personalized medicine, which aims to tailor medical interventions based on individual patient characteristics, has gained prominence in the search for more effective and targeted treatments. Within this context, three-dimensional (3D) cell culture models, particularly spheroids, have emerged as valuable tools that bridge the gap between traditional two-dimensional (2D) cell cultures and in vivo studies. Spheroids, characterized by their spherically shaped cellular aggregates, exhibit enhanced physiological relevance by recreating the cell–cell and cell–matrix interactions found in native tissues. This article aims to explore the developing field of spheroid research, focusing on their potential applications in personalized medicine and elucidating the methodologies employed for spheroid seeding.

Traditional 2D monolayer culture cell culture systems are often inadequate in reproducing the complexity of in vivo tissue structures and functions. Spheroids, formed through the self-assembly of cells, provide a closer representation of the native tissue microenvironment. This three-dimensional architecture facilitates cell–cell interactions, nutrient gradients, and spatial organization, closely mimicking the in vivo conditions. As a result, spheroids have gained prominence as an advanced in vitro model that bridges the gap between conventional cell cultures and in vivo studies.

This systematic review aims to provide a comprehensive overview of the methodologies used for spheroid formation, with a particular focus on seeding techniques. Furthermore, it explores the types of cancers most represented by spheroid models and evaluates their potential utility in personalized medicine.

### 1.1. Models for In Vitro Studies

The concept of growing cells outside of the human body began in the early 20th century. In 1907, Ross Harrison obtained the first successful tissue culture by growing nerve fibers from frog embryos, which paved the way for cell culture techniques that have since become fundamental in biomedical research [1].

Traditional 2D monolayer cultures, such as those using HeLa cells established by George Otto Gey in 1951, have served as the fundamental for biomedical and cancer research [2]. The 2D monolayer culture technique dominated in vitro cancer research for much of the 20th century [3]. These models have enabled landmark discoveries in cancer biology, including the identification of oncogenic pathways and initial therapeutic screens. They have been the cornerstone of in vitro cancer research for decades due to their simplicity, cost-effectiveness, and adaptability to high-throughput screening. In these systems, cancer cells are grown on flat, rigid substrates, providing uniform exposure to nutrients, oxygen, and drugs. The standardized conditions in 2D cultures enable reproducible experiments and straightforward readouts, making them ideal for initial mechanistic studies and large-scale drug screening [4]. However, they fall short in replicating the complex microenvironments of in vivo tissues. Cells cultured in monolayer lack the spatial organization, oxygen and nutrient gradients, and dynamic cell–cell interactions characteristic of native tissues. Following isolation from the tissue and the subsequent transfer to 2D culture conditions, both cell morphology and division patterns are altered. Additionally, 2D culturing contributes to the loss of phenotypic diversity [5].

To overcome these limitations, the field has increasingly embraced 3D models, such as spheroids, organoids, and organ-on-chip platforms. Scientific reviews confirm that these models better simulate cell–cell and cell–matrix interactions, as well as the nutrient and oxygen gradients found in actual tumors. This advancement has significantly improved the study of tumor biology and drug resistance [6,7].

Each model possesses distinct advantages and limitations, and their suitability can vary depending on the experimental objectives. Given these differences, a critical evaluation of each model’s capabilities is essential. The choice between 2D and 3D systems should be guided by the biological question being addressed, the complexity of the experimental setup, and the translational relevance of the findings. Table 1 outlines the advantages, disadvantages, and future perspective for both 2D and 3D models. Moreover, Table 1 overviews the contexts in which each model may be most appropriately applied, emphasizing the importance of strategic model selection tailored to both research goals and practical considerations.

Two-dimensional and three-dimensional models are increasingly used in combination—either as an integrated approach or in parallel—each offering complementary advantages. While 2D cultures remain vital for high-throughput and preliminary screening due to their simplicity and reproducibility, 3D models better mimic tumor physiology, making them valuable for studying progression, metastasis, and therapy resistance. Recent advancements in in vitro modeling have extended beyond traditional 2D cultures toward more sophisticated 3D systems and hybrid approaches, aiming to better recapitulate the structural and biochemical complexity of native tissues [18]. For instance, hydrogel-based scaffolds with engineered microenvironments represent a promising direction in bridging the gap [19]. Moreover, hybrid systems bridge the gap between in vitro and in vivo conditions, combining the practicality of 2D with the biological relevance of 3D. This integrative approach strengthens cancer research, drug discovery, and personalized medicine [9,10,11].

However, there is an emerging consensus that research should transition from reliance on traditional 2D models to the adoption of more physiologically relevant 3D models [8,9].

### 1.2. Spheroids in Personalized Medicine

Within the broad category of 3D in vitro models, self-assembling cellular systems such as spheroids and organoids represent a distinct subgroup. These systems are generated through intrinsic cellular organization processes and represent a unique approach to mimicking physiological conditions. While both systems support cell-cell and cell-matrix interactions in a 3D environment, they differ significantly in terms of cellular complexity, structural organization, and functional potential.

Spheroids, first described in the 1970s, are multicellular aggregates that self-assemble into spherical structures [20]. They are typically composed of a single cell type or a limited number of cell types, often derived from cancer cell lines, stem cells, or primary tissues. Due to their relative simplicity, spheroids are primarily used for modeling tumor microenvironments, studying cancer biology, and evaluating drug penetration and cytotoxicity. They lack tissue-specific architecture and functional heterogeneity characteristic of native organs [21].

In contrast, organoids are cell-derived 3D structures that represent key structural and functional features of their tissue of origin. Generated from stem cells, organoids undergo self-organization and differentiation process that generate multiple, lineage-specific cell types arranged in a physiologically relevant architecture. As a result, organoids closely mimic the in vivo organization and function of organs such as the intestine, brain, kidney or liver [21].

Personalized medicine aims to customize medical interventions based on individual patient characteristics, and spheroids emerge as a powerful tool in this area. The inherent heterogeneity within patient populations can be more accurately represented using spheroids, enabling the development of personalized therapeutic strategies. By incorporating patient-specific cells into spheroid models, researchers can assess drug responses and tailor treatment regimens for improved clinical outcomes [22].

Spheroids have emerged as a pivotal three-dimensional (3D) in vitro model in cancer research, offering a significant leap forward in replicating the complexity of in vivo tumor biology. This enhanced physiological relevance makes spheroids an invaluable tool for advancing our understanding of cancer biology and improving preclinical evaluations of therapeutic agents [21,23].

The defining characteristic of spheroids is their ability to develop internal gradients of oxygen, carbon dioxide, nutrients, and metabolites. These gradients lead to the development of distinct cellular zones within the spheroid [21,23].

Cellular zones of spheroids are visualized and presented in Figure 1. Three cellular zones of spheroids are as follows:Proliferative outer layer: Consisting of actively dividing cells, with high accessibility to oxygen and nutrients.Quiescent intermediate layer: Consisting of quiescent and senescent cells with reduced metabolic activity due to limited nutrient and oxygen availability.Hypoxic apoptotic core: Consisting of cells in an apoptotic state due to severe nutrient and oxygen deprivation. Core environment mimics what is observed in poorly vascularized tumor regions in vivo. Presence of hypoxic core depends on spheroid size, nutrient availability and culture environment.

This zonal architecture replicates the heterogeneous microenvironment of solid tumors, which is critical for studying tumor progression, metastasis, and resistance to therapies.

Spheroids are widely utilized in cancer research as robust models for examining critical biological processes in a three-dimensional context. They provide significant insights into tumor progression by enabling the detailed investigation of invasion, metastasis, and the interactions between the tumor and surrounding stroma. In the field of therapeutic screening, spheroids facilitate the evaluation of anticancer drug efficacy, penetration dynamics within the tumor microenvironment, and mechanisms of drug resistance. They are also essential for modeling tumor responses to hypoxia and radiation therapy. Additionally, spheroids play a significant role in immunotherapy research by supporting the study of immune cell–tumor interactions, thereby providing a more representative alternative to conventional two-dimensional culture models [22,23].

## 2. Materials and Methods

A comprehensive search was conducted in the PubMed database to identify relevant articles available until December 2024. The search utilized specific free words and Medical Subject Headings (MeSH) terms, including key terms such as ‘spheroid,’ ‘cancer,’ ‘drug,’ ‘patient-derived,’ and ‘tumor.’ The exclusion terms ‘co-culture’ and ‘xenograft’ were applied. Only articles published in English were considered for inclusion.

Inclusion criteria encompassed studies involving spheroids formed from patient-derived cancer cells, while exclusion criteria covered articles that did not meet these inclusion criteria such as reviews, case reports, and works focusing on the application or use of spheroids in areas other than drug screening.

Data extraction from the included articles comprised information on authors, publication year, title, cancer type, cell aggregation protocol.

The inclusion criterion of English-language publications may have introduced potential bias, excluding the relevant literature in other languages. Consequently, the findings should be interpreted cautiously, acknowledging the potential limitations associated with the language-based selection criteria.

## 3. Results and Discussion

### 3.1. Database Screening

Of the 190 articles retrieved from PubMed on spheroid seeding from human cancer cells for drug sensitivity screening, only 143 articles were evaluated for eligibility based on the search strategy outlined in the Materials and Methods section. The selection process is illustrated in Figure 2. The first step involved screening the PubMed database using specific keywords and exclusion terms, which yielded 190 articles. Of these, 47 were excluded based on predefined criteria. A total of 143 original, English-language studies conducted on human samples or human cancer cell lines were included.

### 3.2. Systematic Review

Table 2 displays the selected 143 articles along with their fundamental details, including the following: first author and publication year, cancer type by type of tissue in which cancer originates (histological type) and by primary site, and spheroid seeding method.

#### 3.2.1. Source of Spheroids

Spheroids derived from various cancer types are extensively utilized in research to mimic in vivo tumor characteristics, providing insights into diverse cancer-specific processes. The ability to generate spheroids from a variety of cell sources, including patient-derived tumor cells, further enhances their value in studying personalized therapeutic responses. In the present systematic review, the most common cell sources of spheroids were identified. Figure 3 illustrates the distribution of the included studies based on the tissue origin of the spheroid cell sources. The most frequently represented cancer type was breast cancer, accounting for 46 studies. This was followed by colon (23 studies), lung (20 studies), and brain (17 studies). Ovarian cancer spheroids were formed in 17 studies. Other cancer types, such as liver, cervix, pancreas, prostate, and sarcoma, were represented in fewer than 15 studies each. Less commonly studied sources included gastric, bladder, kidney, skin, bone, and thyroid tissues, each with fewer than six studies. This distribution highlights a predominance of spheroid models derived from breast and gastrointestinal cancers, suggesting a focus on these tumor types in current 3D culture-based research.

Table 3 summarizes that the five most investigated cancer origins associated with spheroids are breast (*n* = 46, accounting for 24.6% of all investigated cancers), followed by colon (*n* = 23, 12.3%), lung (*n* = 21, 10.7%), ovary (*n* = 19, 9.6%), and brain (*n* = 18, 9.6%).

##### Breast Cancer

Spheroids derived from breast cancer cells represent one of the most extensively studied in vitro models, reflecting the prominence of breast cancer as both a clinical challenge and a leading research focus. A systematic review of the literature revealed that breast cancer cell lines and patient-derived cells were the most frequently used sources for spheroid generation.

Breast cancer spheroid models facilitate the exploration of unique tumor characteristics (e.g., variations in growth dynamics, gene expression profiles, and interactions with the tumor microenvironment) and aspects of cancer biology (e.g., immortality, telomerase activation, antiapoptotic strategy) [30,38,58,167].

Moreover, this approach facilitates the exploration of therapies targeting estrogen-metabolizing enzymes and receptors, enabling the discovery of novel treatments that may prevent tumor initiation or inhibit cancer growth [36].

Additionally, this methodology supports the development of patient-specific drugs, thereby aligning with the principles of precision medicine to optimize therapeutic outcomes for individual breast cancer patients [69].

##### Colon Cancer

Colon cancer is the second most common source of cells used for spheroid generation, reflecting its critical role in cancer research.

To enhance the utility of human 3D colorectal cancer spheroid models in preclinical drug assessments, there is a need for standardized and validated methodologies. While monoculture spheroids are useful for high-throughput drug screening due to their simplicity, spheroids provide deeper insights into tumor biology and chemoresistance mechanisms, offering a more accurate preclinical tool for evaluating therapeutic efficacy and developing new drug candidates [42,168].

The 3D cultures still face challenges in clinical implementation, and advancements in co-culture techniques, addressing tumor heterogeneity, and improving laboratory protocols are essential for enhancing reproducibility and drug testing reliability in colorectal cancer research [169]. Moreover, collecting cells during biopsy from different tumor sites might provide a more comprehensive representation of tumor subclones, offering greater insight into the tumor’s diverse properties and improving the accuracy of preclinical models for drug testing and personalized medicine [169].

##### Lung Cancer

Lung cancer represents the third most common source of cells for generating spheroids. Lung cancer-derived spheroids are utilized to investigate key aspects of tumor biology.

Lung cancer spheroids are particularly valuable for studying the progression of both non-small-cell lung cancer (NSCLC) and small-cell lung cancer (SCLC), which differ significantly in their biological behavior and response to therapy [170,171].

Therapeutically, lung cancer spheroids serve as platforms for evaluating the efficacy of novel anticancer agents, including small molecules, biologics, and combination therapies. Their three-dimensional structure facilitates studies of drug delivery systems aimed at overcoming barriers such as limited penetration into solid tumors [50,57,59,77,87,129].

##### Ovarian Cancer

Ovarian cancer is the fourth most common source of cells used for spheroid generation. Spheroids derived from ovarian cancer cells are utilized in research to study the processes central to ovarian cancer pathology, such as peritoneal metastasis, chemoresistance, and interactions with the tumor microenvironment. Given the propensity of ovarian cancer to spread via the peritoneal cavity through multicellular aggregates, spheroids serve as a physiologically relevant model to replicate these metastatic behaviors in vitro [41,62,86,116,117].

Moreover, patient-derived ovarian cancer spheroids are increasingly used for precision oncology, allowing for the evaluation of personalized therapeutic strategies tailored to the molecular profiles of individual tumors [123,159].

##### Brain Cancer

Brain cancers, including glioblastoma and other gliomas, rank as the fifth most common source of cells used for spheroid generation.

These three-dimensional models are crucial for studying the unique microenvironment and invasive properties of brain tumors, which are characterized by their aggressive behavior and resistance to standard therapies. Brain cancer-derived spheroids closely mimic the in vivo conditions of brain tumors, providing insights into key processes such as tumor invasion, therapeutic resistance, and interactions with the extracellular matrix (ECM) [73,84,103]. Glioblastoma-derived spheroids are among the most studied in this category. They are particularly valuable for investigating the highly invasive nature of glioblastoma cells, which infiltrate the surrounding healthy brain tissue, making complete surgical resection nearly impossible [34,114].

Recent advancements include patient-derived brain cancer spheroids, which preserve the genetic and phenotypic heterogeneity of primary tumors. These models are increasingly used for personalized medicine, enabling the testing of individualized therapeutic regimens in a controlled in vitro setting [30,73,157].

#### 3.2.2. Spheroid Seeding Methods

The successful implementation of spheroids in personalized medicine relies on robust and reproducible methodologies for their generation. The generation of three-dimensional (3D) spheroids as in vitro models requires the careful consideration of seeding methods to ensure reproducibility, scalability, and physiological relevance. A variety of techniques have been developed to create spheroids, ranging from traditional methods to advanced approaches incorporating cutting-edge technologies. 

In conclusion, this article aimed to provide a thorough understanding of the methodologies employed in spheroid seeding and highlighted the manifold applications of spheroids in advancing personalized medicine.

Figure 4 presents the distribution of the spheroid seeding methods utilized across the included studies. The most employed approach was the forced-floating method, with over 70 studies utilizing various subtypes of this technique. Among these, ultra-low attachment (ULA) plates were the most frequently used, followed by poly-HEMA-coated surfaces, agar-coated wells, liquid overlay, and other less common variations. Scaffold-based methods were also widely applied, appearing in 40 studies. In contrast, the hanging drop technique and suspension culture were used less frequently, with approximately 13 and 4 studies, respectively. A smaller but notable number of studies employed recently developed or advanced methods, indicating ongoing innovation in spheroid formation strategies. This distribution underscored the dominance of forced-floating techniques in current spheroid culture protocols, while also reflecting methodological diversity and emerging alternatives.

Table 4 summarizes the five most investigated spheroid seeding methodologies; the most common were forced-floating (*n* = 70, accounting for 31.8% of studies), scaffold-based methods (*n* = 41, 18.6%), recent advances (*n* = 21, 9.5%), hanging drop (*n* = 14, 6.4%) and suspension culture (*n* = 4, 1.8%). A summary of the systematic review results is presented.

Based on the systematic review, a summary of the most common spheroid seeding methods is illustrated in Figure 5. The figure visualizes the most utilized spheroid seeding methods such as hanging drop (1), forced-floating (2), magnetic levitation (3), scaffold-based (4), suspension culture (5), and recent scientific advances (6) including: microencapsulation (6a), bioprinting (6b), nanoparticle-assisted techniques (6c), microfluidics (6d) and lab-on-a-chip (6e).

##### Forced-Floating

The forced-floating method is the most employed technique for spheroid formation due to its simplicity and scalability. This approach uses non-adhesive surfaces to prevent cell attachment, encouraging cells to aggregate and form three-dimensional (3D) spheroids. The technique involves seeding cells in multi-well plates that have been treated to inhibit surface adhesion, either through coating with low-attachment materials like poly-HEMA (Poly(2-hydroxyethyl methacrylate)) or using ultra-low attachment (ULA) culture plates designed specifically for this purpose. In the absence of adhesion sites, cells naturally aggregate in the medium, forming spheroids under the influence of gravity and intercellular interactions. This process begins with the preparation of a single-cell suspension of the desired cell density, typically ranging from 10^3^ to 10^5^ cells per well, depending on the type of cells and the intended spheroid size. The cell suspension is then distributed into the wells of the plate. After 24–96 h, depending on the cell type and experimental conditions, the cells self-assemble into compact spheroids [31,109,148,163].

The forced-floating method is particularly advantageous for producing uniform spheroids with consistent size and morphology, making it suitable for high-throughput applications such as drug screening and toxicity assays [11,39,63,65,100,148]. Additionally, this method does not require specialized equipment beyond the plates or coatings, making it accessible for most laboratories. One of the significant benefits of the forced-floating method is its compatibility with automated systems, allowing for large-scale spheroid generation and analysis. [11,63,148]. However, the method has certain limitations. The reliance on non-adhesive surfaces can lead to variability in spheroid integrity and size if the cell density or culture conditions are not carefully optimized [11,63]. Additionally, long-term culture may be constrained by limited nutrient and oxygen diffusion, necessitating periodic medium exchange or supplementation with perfusion systems [65].

##### Scaffold-Based

The scaffold-based method for spheroid seeding is the second most utilized method for generating three-dimensional (3D) cellular aggregates. This method uses biomaterials, known as scaffolds, that mimic the extracellular matrix (ECM) to provide the structural support and environment conducive to cell adhesion, proliferation, and aggregation into spheroids. Scaffolds can be fabricated from a wide range of materials, including natural polymers such as collagen, gelatin, and alginate, as well as synthetic polymers like PLGA (poly(lactic-co-glycolic acid)) and PEG (polyethylene glycol). The process typically begins with the preparation of the scaffold material, which may be in the form of hydrogels, porous matrices, or microcarriers. Cells are then seeded onto or encapsulated within the scaffold. Once seeded, the cells interact with the scaffold material and with one another, eventually forming spheroid structures. The scaffold not only facilitates cell aggregation, but it also supports nutrient and oxygen diffusion, which is crucial for maintaining cell viability and function in 3D cultures [15,148,172].

Scaffold-based methods offer significant advantages, including the ability to recreate a more physiologically relevant microenvironment compared to non-adhesive-based techniques. The structural and biochemical properties of the scaffold can be engineered to closely mimic the in vivo ECM, supporting the growth and differentiation of specific cell types [15]. This makes the method particularly suitable for modeling complex tissues and for co-culture systems involving multiple cell types [173]. Additionally, scaffold-based systems are compatible with long-term culture, as the scaffold provides a sustained environment for nutrient and waste exchange [15]. However, there are limitations to this approach. The use of scaffolds introduces variability in spheroid size and shape, depending on the uniformity of the material and the seeding protocol [15]. The composition and mechanical properties of the scaffold can also influence cellular behavior, which may complicate the interpretation of results [15]. Furthermore, the cost and complexity of scaffold fabrication, particularly for advanced synthetic materials, may be a barrier for some applications [11,15].

##### Hanging Drop

The hanging drop method is robust and is the third most utilized technique for generating spheroids in vitro, particularly valued for its ability to produce uniform and physiologically relevant cellular aggregates. This method capitalizes on gravity-driven cellular self-assembly within droplets of culture medium, facilitating interactions that mimic those found in vivo. To implement this approach, a cell suspension of the desired density is prepared, often ranging from 10^2^ to 10^4^ cells per droplet. Droplets, typically 20–50 μL in volume, are then dispensed onto the inner surface of an inverted Petri dish lid. The droplets are retained by surface tension, allowing them to remain suspended. This setup is placed over a dish containing a hydrating agent, such as phosphate-buffered saline (PBS) or water, to maintain humidity and prevent evaporation during incubation. After 24 to 72 h under standard culture conditions, the suspended cells settle at the bottom of the droplets and aggregate into spheroids through intercellular adhesion and natural cell–cell interactions [24,41,60,120,122,151].

The hanging drop method is particularly advantageous due to its simplicity, low cost, and minimal equipment requirements [11,60,174]. This method offers significant benefits, including enhanced cellular aggregation and adhesion driven by gravitational forces, which minimize mechanical damage to spheroids. It also allows for precise control over the spheroid size and cell composition by adjusting the initial cell density and droplet volume [11,60]. However, this technique has limitations, such as the potential disruption of spheroids during transfer to conventional culture plates due to mechanical stress. Moreover, it is labor-intensive and not easily scalable for high-throughput applications, as each droplet must be individually prepared and managed. Additionally, nutrient and waste exchange are limited by the small volume of medium, necessitating careful monitoring to maintain spheroid viability [11,120,140,175].

##### Suspension Culture

The suspension culture method is the fourth most utilized approach for generating spheroids in in vitro studies, particularly in cancer research, developmental biology, and drug discovery. This technique relies on culturing cells in a liquid medium without a solid substrate, allowing them to aggregate and form spheroids due to intercellular adhesion and natural aggregation tendencies. Typically, the suspension culture is conducted in culture vessels, such bioreactors or spinner flasks, to prevent cell adhesion to the container surface and promote spheroid formation [148,175]. The process begins with the preparation of a single-cell suspension at a defined density, which is a critical parameter for achieving a uniform spheroid size and morphology. These vessels prevent cells from adhering to the surface and maintain them in suspension. Spinner flasks or bioreactors are dynamic systems, where gentle agitation or rotation keeps the cells suspended and evenly distributed, which can enhance the uniformity of spheroid formation and improve the mass transport of nutrients and oxygen [30,148,175].

The suspension culture method offers several advantages. It is relatively simple and cost-effective, requiring minimal specialized equipment beyond vessels. The method is highly versatile, accommodating various cell types and allowing for the easy incorporation of co-culture systems to model complex cell–cell interactions, such as those between tumor and stromal cells. Furthermore, the suspension culture can be adapted for high-throughput applications, making it suitable for large-scale drug screening and toxicity testing [11,175,176]. Despite its strengths, the suspension culture method has limitations. Nutrient and oxygen diffusion can be inadequate in larger spheroids, leading to hypoxic or necrotic cores. This limitation necessitates the careful control of spheroid size and medium composition to maintain viability. Additionally, while the method is relatively straightforward, achieving consistent spheroid size and morphology can be challenging without precise control over cell seeding density and culture conditions [11,175,176]. Moreover, prolonged culture durations may require frequent medium changes or the use of perfusion systems to sustain spheroid health [11,175,176].

##### Magnetic Levitation

The magnetic levitation method is a less frequently used approach to spheroid formation. This method utilizes leveraging magnetic fields to promote cellular aggregation and 3D structure development. This technique involves the use of magnetic nanoparticles that are internalized by cells through incubation. The nanoparticles are typically composed of biocompatible materials such as iron oxide and may be functionalized with ECM proteins or other molecules to enhance cellular uptake and minimize toxicity. After the cells have internalized the nanoparticles, they are exposed to a magnetic field, which forces them to aggregate and suspend in the culture medium. The magnetic field enables the controlled formation of compact spheroids by facilitating cell–cell and cell–ECM interactions [175,177]. Magnetic levitation offers several advantages. It allows for rapid and reproducible spheroid generation and provides a high degree of control over spheroid size and structure. Additionally, this method supports the formation of co-culture spheroids by enabling the simultaneous aggregation of different cell types, which is particularly useful for modeling tumor–stroma or tumor–immune cell interactions. The technique also facilitates the incorporation of ECM components, improving the physiological relevance of the spheroid microenvironment. Furthermore, magnetic levitation is amenable to high-throughput applications and can be easily scaled up for drug screening or other large-scale studies. Despite its advantages, the method has limitations. The requirement for magnetic nanoparticles introduces potential concerns regarding biocompatibility and cellular toxicity, particularly for long-term studies. Additionally, nanoparticle uptake among cells can vary, potentially leading to inconsistencies in spheroid formation. The cost of magnetic nanoparticles and specialized magnetic devices may also be a barrier for some laboratories [11,175,177].

##### Recent Advances in Spheroid Seeding Methods

Recent advances in spheroid seeding include novel techniques that have emerged in the past few years and do not fall within the conventional classifications of standard seeding methods. These approaches represent hybrid or innovative strategies that extend beyond traditional categories such as hanging drop or low-attachment culture. Recent advances in spheroid seeding methods are represented by microencapsulation, bioprinting, nanoparticle-assisted techniques, microfluidics, and lab-on-a-chip methods.

Microencapsulation involves embedding cells or spheroids within biocompatible hydrogels to recreate ECM-like environments. This technique enhances cell–cell interactions, shields cells from shear stress, and supports co-culture configurations. However, traditional methods often suffer from inconsistent spheroid loading, size variability, nutrient diffusion limitations, and difficulties in spheroid retrieval [165,178]. For example, Chan et al. developed a microfluidic double-emulsion system that directly generates uniform microencapsulated hepatocyte spheroids (<200 μm) within 4 h. Using an alginate–collagen hydrogel matrix, the encapsulated spheroids exhibited superior hepatic functions—albumin and urea secretion, and cytochrome P450 activity—over 24 days compared to alginate-only or collagen-sandwich cultures [179].

Bioprinting is a method utilizing 3D printing technologies, allowing for the precise placement of cells and biomaterials to replicate native tissue architecture. It offers exceptional control over spheroid organization and is particularly effective for high-throughput applications and heterotypic co-cultures. Limitations include the need for bioink optimization and the high costs of equipment. Bioprinting enables the spatially controlled deposition of cell-laden bioinks to organize spheroids into functional 3D tissues [180,181]. Extrusion-based bioprinting is an effective method for spheroid seeding, that fabricates tissue constructs by continuously dispensing bioink through a nozzle, allowing for the precise deposition of cell-laden materials. It is valued for its simplicity, scalability, and affordability, but shear stress during extrusion is a critical factor influencing cell viability and print quality [182]. Inkjet is a non-contact, droplet-based technique that deposits picolitre-sized droplets of bioink—comprising cells and biomaterials—onto substrates using thermal or piezoelectric actuation. This method offers high-resolution patterning and cost-effectiveness but is constrained by the requirement for low-viscosity bioinks, limiting the achievable cell densities and construct complexity [183,184].

Nanoparticle-assisted methods refer to a broader set—than magnetic levitation—of strategies in which nanoparticles are used to support or enhance various aspects of spheroid formation, without necessarily involving magnetic fields or levitation. This method precisely guides cell aggregation into spheroids by leveraging functionalized nanoparticles. Nanoparticles—ranging from magnetic to non-magnetic (e.g., gold, silica, polymer-based)—serve diverse roles such as carriers for bioactive agents, enhancers of cell adhesion, or modulators of the microenvironment. Unlike magnetic levitation, these techniques do not directly position or aggregate cells but rather provide biochemical or structural cues that facilitate spheroid development, improve viability, or guide differentiation. This method is highly versatile and can be tailored to specific experimental goals, including drug testing or tissue engineering. It offers reproducibility and integration with imaging or therapeutic applications but raises concerns about nanoparticle cytotoxicity and scalability [46,73,138,179].

Microfluidic and lab-on-a-chip are technologies of spheroid seeding that offer precise control over the microenvironment for spheroid formation, enabling uniformity in size and structure, and facilitating high-throughput applications [35,41,57,90,94,108,185]. An example of a microfluidic platform might be a microwell-based system. These systems utilize arrays of microwells to confine cells, promoting self-aggregation into spheroids. For instance, pyramidal microwells with a 90° tip angle have been shown to enhance the spheroid formation and cardiac differentiation of mouse embryonic stem cells [186].

##### Summary of Spheroid Seeding Method

A summary of the advantages and limitations of selected spheroid seeding methods is presented in Table 5.

## 4. Conclusions

This systematic review underscores the expanding role of three-dimensional (3D) cancer cell models, particularly spheroids, in advancing cancer research, drug development, and personalized medicine. Among the studies analyzed, breast cancer was the most frequently investigated source for spheroid formation, followed by colon, lung, ovarian, and brain cancers, reflecting their clinical relevance and research priority. Forced-floating and scaffold-based techniques emerged as the most employed methods due to their relative simplicity, reproducibility, and broad applicability across various cancer types. The hanging drop technique, although more labor-intensive, is gaining traction for its ability to produce physiologically relevant 3D structures.

Despite the progress made, establishing reproducible protocols tailored to different spheroid seeding techniques remains a significant challenge, limiting efforts toward standardization and the broader adoption of these models in therapeutic screening. Additionally, conventional spheroid models often fall short in replicating the structural and functional complexity of in vivo tumors, particularly with respect to tumor heterogeneity and microenvironmental interactions. The increasing application of patient-derived spheroids represents a promising development, offering models that better capture the genetic and phenotypic variability of individual tumors and supporting efforts toward more personalized therapeutic strategies.

### Future Perspectives

Future research should prioritize the development of standardized and reproducible protocols for spheroid generation, especially those compatible with high-throughput drug screening platforms. Integrating tumor heterogeneity—with patient-derived models and sampling from multiple tumor regions—will be essential to improve model relevance and predictive accuracy. Additionally, incorporating co-culture systems that include stromal, immune, and vascular components may better replicate the tumor microenvironment and enhance the physiological fidelity of in vitro models. Advancements in the automation, miniaturization, and real-time monitoring of spheroid cultures will further support the translation of 3D cancer models into clinically meaningful applications, ultimately contributing to the development of more effective, patient-tailored cancer therapies.

## Figures and Tables

**Figure 1 ijms-26-06478-f001:**
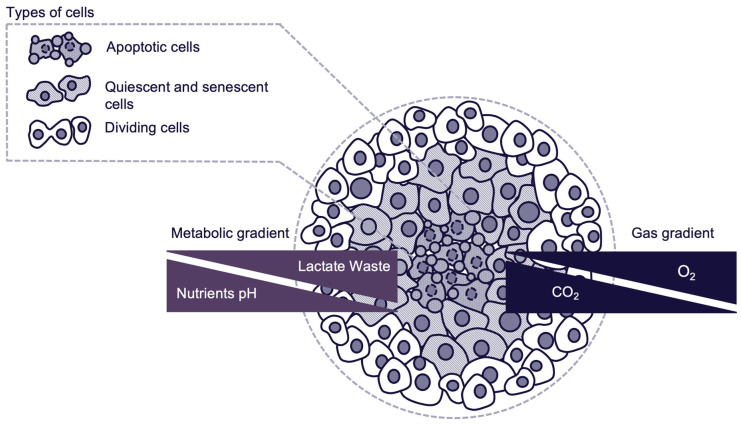
Schematic representation of the cellular structure of a spheroid.

**Figure 2 ijms-26-06478-f002:**
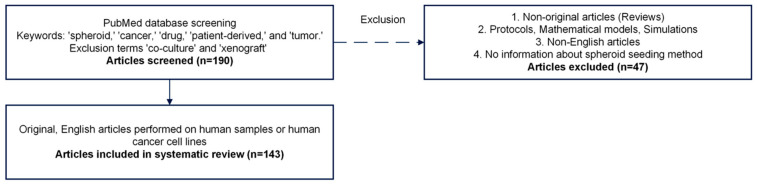
Flowchart of the study selection process for the systematic review, including inclusion and exclusion criteria, and the final number of studies selected.

**Figure 3 ijms-26-06478-f003:**
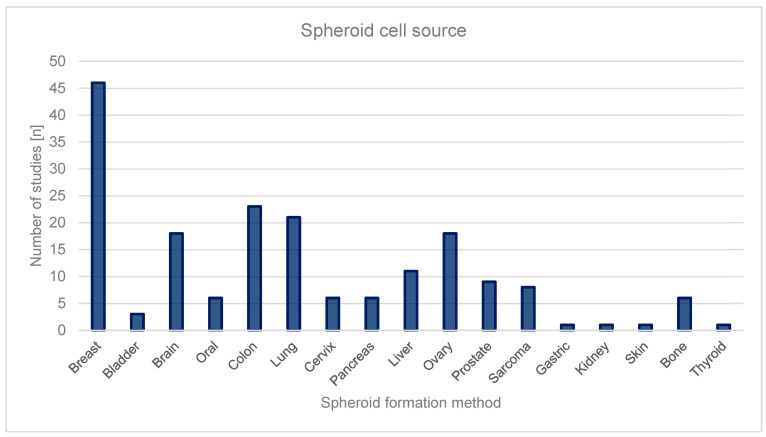
Distribution of cell sources used for spheroid generation presented as a column graph. The figure summarizes the number of studies reporting each cell source, as identified though the systematic review.

**Figure 4 ijms-26-06478-f004:**
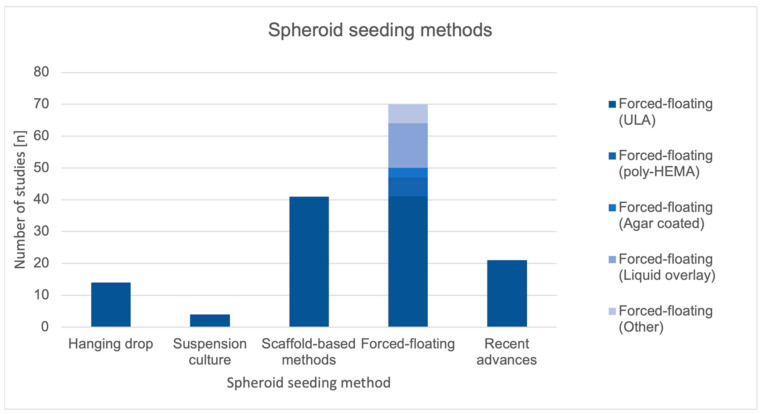
Distribution of spheroid seeding methods used for spheroid generation presented as a column graph. The figure summarizes the number of studies reporting each seeding method, as identified though the systematic review.

**Figure 5 ijms-26-06478-f005:**
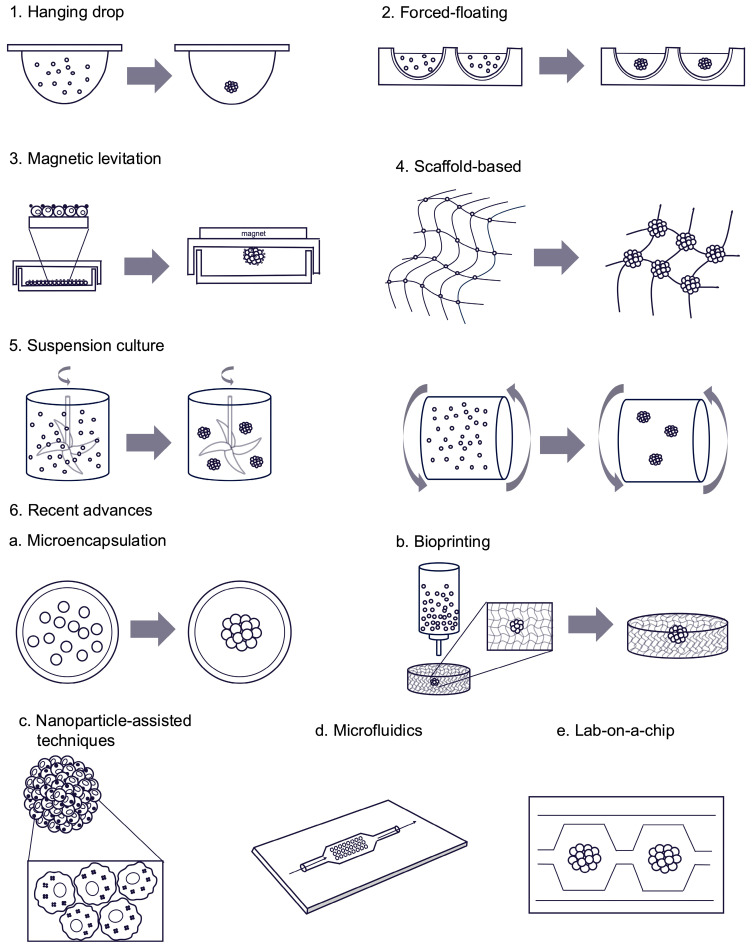
Overview of techniques used for spheroid formation. The figure provides a graphical representation of the various seeding methods employed in the generation of spheroids, as identified in the systematic review.

**Table 1 ijms-26-06478-t001:** Advantages, limitations, and potentials of 2D and 3D models.

	2D Models	3D Models
Advantages	Simplicity and ease of use [5,8]. Cost-effectiveness and availability [5,9,10]. High-throughput capability [7,8].	Closer mimicry of in vivo conditions [10]. Replication of tumor microenvironment and cellular interactions (e.g., cell–cell and cell–matrix interactions, hypoxic core) [7,8,10,11,12,13]. Better simulation of drug penetration and resistance (drug diffusion barriers and heterogeneous cellular responses) [7,8,10,11,12]. Representation of gene and protein expression profile that reflects tumors [10,12,14].
Limitations	Lack of physiological relevance [8,11,13]. Limited replication of cell–cell and cell–matrix interactions [10,11,13]. Fails to simulate gradients of oxygen, nutrients, and metabolites [5].	Higher cost and technical complexity [5,9,10,12]. Longer culture time [5]. Scalability challenges [15]. Reproducibility issues in some systems [5,9].
Potentials	Effective for initial drug screening and basic mechanistic studies [12,15]. Suitable for large-scale studies with consistent and reproducible outputs [12].	Ideal for studying tumor progression, metastasis, and resistance mechanisms [11,16]. Promising for personalized medicine applications, including patient-derived models [17].

**Table 2 ijms-26-06478-t002:** Articles selected for systematic review.

Reference	Cancer Primary Site (Histological Type)	Cell Line	Spheroid Formation Method
Agastin S, 2011 [24]	Colon (adenocarcinoma), Breast (adenocarcinoma)	Colo205, MDA-MB-231	Hanging drop
Alhasan L, 2016 [25]	Breast (carcinoma)	BT-474	Scaffold-based methods
An HJ, 2020 [26]	Kidney (carcinoma)	A498	Scaffold-based methods
Árnadóttir SS, 2018 [27]	Colon	Patient-derived	Forced-floating (not defined)
Baek N, 2016 [28]	Prostate (carcinoma), Bone (neuroblastoma), Lung (carcinoma), Cervix (adenocarcinoma), Bone (osteosarcoma)	DU145, SH-SY5Y, A549, HeLa, HEp2, 0U-2OS	Forced-floating (agar-coated plates)
Barone RM, 1981 [29]	Colon (adenocarcinoma)	HT-29	Suspension culture
Bartholomä P, 2005 [30]	Breast (carcinoma)	T-47D	Suspension culture
Boo L, 2020 [31]	Breast (adenocarcinoma)	MCF-7	Forced-floating (agar-coated plates)
Brooks EA, 2019 [32]	Ovary (adenocarcinoma)	Patient-derived, AU565, BT549, SKOV-3	Scaffold-based methods
Bruns J, 2022 [33]	Brain (glioblastoma)	Patient-derived, U87	Scaffold-based methods
Calori IR, 2022 [34]	Brain (glioblastoma, medulloblastoma)	U87, T98G, A172, UW473	Forced-floating (ultra-low attachment (ULA))
Chang S, 2022 [35]	Breast (adenocarcinoma)	MCF-7	Recent advances
Chen G, 2022 [36]	Breast (adenocarcinoma)	MCF-7	Forced-floating (ULA)
Chen MC, 2010 [37]	Breast (melanoma)	LCC6/Her-2	Recent advances
Chen Z, 2021 [38]	Breast (adenocarcinoma)	MDA-MB-231	Forced-floating (ULA)
Cheng V, 2015 [39]	Brain (glioblastoma)	U87, U251	Forced-floating (ULA)
Close DA, 2022 [40]	Oral (squamous cell carcinoma)	Cal33, FaDu, UM-22B, OSC-19	Forced-floating (ULA)
Das T, 2013 [41]	Ovary (adenocarcinoma)	TOV112D	Hanging drop
Das V, 2016 [42]	Colon (carcinoma), Colon (adenocarcinoma), Bone (osteosarcoma), Cervix (adenocarcinoma), Colon (adenocarcinoma), Liver (carcinoma)	HCT116, HT29, U-2OS, HeLa, Caco-2, HepG2	Forced-floating (liquid overlay)
Das V, 2017 [43]	Colon (carcinoma)	HCT116	Forced-floating (liquid overlay)
De Angelis ML, 2018 [44]	Colon	Patient-derived	Forced-floating (ULA)
Dhamecha D, 2021 [45]	Lung (carcinoma), Bone (osteosarcoma)	A549, MG-63	Scaffold-based methods
Dias DR, 2016 [46]	Cervix (adenocarcinoma)	HeLa	Recent advances
Domenici G, 2021 [47]	Bone (sarcoma)	Patient-derived	Forced-floating (ULA)
Dufau I, 2012 [48]	Pancreas (adenocarcinoma)	Capan-2	Forced-floating (Poly(2-hydroxyethyl methacrylate) (poly-HEMA))
Eetezadi S, 2018 [49]	Ovary (carcinoma), Ovary (adenocarcinoma)	UWB1.289, UWB1.289+BRCA1, OV-90, SKOV3, PEO1, PEO4, COV362	Forced-floating (ULA)
Eguchi H, 2022 [50]	Lung (carcinoma)	A549	Forced-floating (ULA)
Eimer S, 2012 [51]	Brain (glioblastoma)	Patient-derived	Forced-floating (ULA)
El-Sadek IA, 2021 [52]	Breast (adenocarcinoma)	MCF-7	Forced-floating (ULA)
Enmon RM Jr, 2001 [53]	Prostate (carcinoma)	DU 145	Forced-floating (agar plates)
Flørenes VA, 2019 [54]	Skin (melanoma)	Patient-derived	Forced-floating (ULA)
Fu J, 2020 [55]	Liver (carcinoma), Prostate (carcinoma), Lung (carcinoma), Breast (adenocarcinoma)	HepG2, DU 145, A549, MCF-7, MDA-MB-231	Scaffold-based methods
Fu JJ, 2018 [56]	Prostate (carcinoma)	DU 145, LNCap	Scaffold-based methods
Gao Y, 2022 [57]	Lung (carcinoma)	A549	Recent advances
Gencoglu MF, 2018 [58]	Breast (adenocarcinoma), Breast (carcinoma), Prostate (carcinoma), Prostate (adenocarcinoma), Ovary (adenocarcinoma)	AU565, BT549, BT474, HCC 1419, HCC 1428, HCC 1806, HCC 1954, HCC 202, HCC 38, ZR75 1, HCC 70, LNCaPcol, PC3, SKOV3	Scaffold-based methods, Microwells, Suspension culture
Gendre DAJ, 2021 [59]	Lung (mesothelioma), Lung (adenocarcinoma)	H2052, H2052/484, H2452, LuCa1, LuCa61, LuCa62	Scaffold-based methods
Gheytanchi E, 2021 [60]	Colon (adenocarcinoma)	HT-29, Caco-2	Hanging drop, Forced-floating (poly-HEMA)
Goisnard A, 2021 [61]	Breast (carcinoma), Breast (adenocarcinoma)	SUM1315, MDA-MB-231, HCC1937, SW527, DU4475	Forced-floating (ULA)
Guo X, 2019 [62]	Ovary (adenocarcinoma), Colon (adenocarcinoma), Pancreas (carcinoma), Prostate (adenocarcinoma)	OVCAR3, SW620, PANC-1, PC3	Scaffold-based methods
Hagemann J, 2017 [63]	Oral (carcinoma)	FaDu, Cal27, UPCI-SCC-154	Forced-floating (ULA), Hanging drop
Han S, 2022 [64]	Liver	Patient-derived	Forced-floating (ULA)
Harmer J, 2019 [65]	Brain (glioblastoma)	U251, KNS42	Scaffold-based methods
Herter S, 2017 [66]	Colon (adenocarcinoma)	LS174T, LoVo	Hanging drop
Ho WY, 2012 [67]	Breast (adenocarcinoma)	MCF-7	Forced-floating (liquid overlay)
Ho WY, 2021 [68]	Breast (adenocarcinoma)	MCF-7	Scaffold-based methods
Hofmann S, 2022 [69]	Breast	Patient-derived	Forced-floating (ULA)
Hornung A, 2016 [70]	Colon (adenocarcinoma)	HT-29	Scaffold-based methods
Huang Z, 2020 [71]	Breast (adenocarcinoma)	MDA-MB-231	Scaffold-based methods
Jove M, 2019 [72]	Breast (adenocarcinoma), Colorectal (adenocarcinoma)	MCF-7, DLD-1	Scaffold-based methods
Ju FN, 2023 [73]	Brain (glioblastoma)	U87	Recent advances
Karamikamkar S, 2018 [74]	Breast (adenocarcinoma)	MCF-7	Scaffold-based methods
Karlsson H, 2012 [75]	Colon (carcinoma)	HCT-116	Forced-floating (ULA)
Karshieva SS, 2022 [76]	Colon (carcinoma), Liver (carcinoma)	HCT-116, Huh7	Forced-floating (ULA)
Kato EE, 2021 [77]	Lung (carcinoma)	A549	Hanging drop
Kim CH, 2020 [78]	Liver (carcinoma)	HepG2	Recent advances
Ko J, 2019 [79]	Brain (glioblastoma)	U87	Scaffold-based methods
Kochanek SJ, 2019 [80]	Oral (carcinoma)	Cal33, Cal27, FaDu, UM-22B, BICR56, OSC-19, PCI-13, PCI-52, Detroit-562, UM-SCC-1, and SCC-9	Forced-floating (ULA)
Kochanek SJ, 2020 [81]	Oral (carcinoma)	Cal33, FaDu, UM-22B, BICR56, OSC-19	Forced-floating (ULA)
Koshkin V, 2016 [82]	Breast (adenocarcinoma)	MCF-7	Scaffold-based methods
Kroupová J, 2022 [83]	Colon (adenocarcinoma)	HT-29	Forced-floating (not defined)
Kudláčová J, 2020 [84]	Brain (glioblastoma)	U87	Forced-floating (ULA)
Kumari P, 2017 [85]	Cervix (adenocarcinoma), Lung (carcinoma)	HeLa, A549	Scaffold-based methods
Lal-Nag M, 2017 [86]	Ovary (adenocarcinoma)	Hey-A8–GFP	Forced-floating (ULA)
Lama R, 2013 [87]	Lung (carcinoma)	H292	Scaffold-based methods
Landgraf L, 2022 [88]	Prostate (adenocarcinoma), Brain (glioblastoma)	PC-3, U87	Forced-floating (liquid overlay)
Le VM, 2016 [89]	Lung (carcinoma), Colon (carcinoma), Brain (glioblastoma)	95-D, HCT-116, U87	Scaffold-based methods
Lee SW, 2019 [90]	Lung (carcinoma)	A549	Recent advances
Lee Y, 2022 [91]	Lung (carcinoma)	H460, A549	Forced-floating (ULA)
Lemmo S, 2014 [92]	Breast (adenocarcinoma)	MDA-MB-231	Scaffold-based methods
Li M, 2019 [93]	Cervix (carcinoma)	C-33-A, DoTC2 4510	Forced-floating (ULA)
Lim W, 2018 [94]	Colon (carcinoma), Brain (glioblastoma)	HCT-116, U87	Recent advances
Lin ZT, 2021 [95]	Breast (adenocarcinoma)	MDA-MB-436	Scaffold-based methods
Liu X, 2021 [96]	Sarcoma	HS-SY-II	Recent advances
Lorenzo C, 2011 [97]	Pancreas (adenocarcinoma)	Capan-2	Forced-floating (poly-HEMA)
Luan Q, 2022 [98]	Lung (adenocarcinoma), Lung (carcinoma)	HCC4006, H1975, A549	Scaffold-based methods, Forced-floating (ULA)
Madsen NH, 2021 [99]	Breast (adenocarcinoma), Colon (adenocarcinoma), Pancreas (carcinoma)	MCF-7, HT-29, PANC-1, MIA PaCa-2	Forced-floating (ULA)
Marshall SK, 2022 [100]	Bone (osteosarcoma)	MG-63	Forced-floating (ULA)
Maruhashi R, 2018 [101]	Lung (carcinoma)	A549	Forced-floating (ULA)
Melnik D, 2020 [102]	Thyroid (carcinoma)	FTC-133	Suspension culture
Molyneaux K, 2021 [103]	Brain (glioblastoma)	LN229, U87, Gli36	Forced-floating (not defined)
Monazzam A, 2007 [104]	Breast (adenocarcinoma)	MCF-7	Forced-floating (agar plates)
Morimoto T, 2023 [105]	Gastric	Patient-derived	Scaffold-based methods
Mosaad EO, 2018 [106]	Prostate (cancer), Prostate (carcinoma)	C42B, LNCaP	Recent advances
Mueggler A, 2023 [107]	Lung	Patient-derived	Scaffold-based methods
Nashimoto Y, 2020 [108]	Breast (adenocarcinoma)	MCF-7	Recent advances
Nigjeh SE, 2018 [109]	Breast (adenocarcinoma)	MDA-MB-231	Forced-floating (agar plates), Forced-floating (ULA)
Nittayaboon K, 2022 [110]	Colon (carcinoma)	PMF-k014	Forced-floating (poly-HEMA)
Ohya S, 2021 [111]	Prostate (carcinoma)	LNCaP	Forced-floating (ULA)
Oliveira MS, 2016 [112]	Breast (adenocarcinoma), Ovary (adenocarcinoma)	MCF-7/Adr, NCI/Adr	Forced-floating (liquid overlay)
Ono K, 2022 [113]	Oral (carcinoma)	SAS, HSC-3, HSC-4, OSC-19	Forced-floating (ULA)
Pampaloni F, 2017 [114]	Brain (glioblastoma)	U343	Forced-floating (liquid overlay)
Park MC, 2016 [115]	Brain (glioblastoma)	Patient-derived and PC14PE6, PC14PE6_LvBr3, D54, LN428, LN751, U251E4, U87E4, SN-12C, SNU-119, SNU-216, SNU-668, SNU-719, HCC1171, HCC1195, HCC15, HCC1588, HCC2108, HCC44	Forced-floating (not defined)
Pattni BS, 2016 [116]	Ovary (adenocarcinoma)	NCI/ADR-RES	Forced-floating (liquid overlay)
Perche F, 2012 [117]	Ovary (adenocarcinoma)	NCI/ADR-RES	Forced-floating (liquid overlay)
Preda P, 2023 [118]	Breast (adenocarcinoma), Brain (glioblastoma)	MDA-MB-231, U87	Scaffold-based methods
Pulze L, 2020 [119]	Breast (adenocarcinoma)	MCF-7	Forced-floating (ULA)
Raghavan S, 2016 [120]	Breast (adenocarcinoma), Ovary (adenocarcinoma)	MCF-7, OVCAR8	Hanging drop, Forced-floating (liquid overlay)
Raghavan S, 2019 [121]	Ovary (adenocarcinoma)	A2780, OVCAR3	Hanging drop
Ralph ACL, 2020 [122]	Breast (adenocarcinoma), Breast (carcinoma)	MCF-7, MDA-MB-231, T47D	Hanging drop
Roering P, 2022 [123]	Ovary (adenocarcinoma)	Patient-derived, CAOV3, OVCAR8	Forced-floating (ULA)
Roudi R, 2016 [124]	Lung (carcinoma)	A549	Forced-floating (poly-HEMA)
Rouhani M, 2014 [125]	Breast (carcinoma)	T47D	Forced-floating (liquid overlay)
Sakumoto M, 2018 [126]	Sarcoma	Patient-derived	Forced-floating (ULA)
Salehi F, 2020 [127]	Breast (adenocarcinoma), Breast (carcinoma)	MDA-MB-231, T47D, MCF-7	Forced-floating (liquid overlay), Hanging drop
Sambi M, 2020 [128]	Breast (adenocarcinoma)	MDA-MB-231	Scaffold-based methods
Sankar S, 2021 [129]	Lung (carcinoma)	A549	Recent advances
Särchen V, 2022 [130]	Sarcoma	RH30	Forced-floating (ULA)
Sarıyar E, 2023 [131]	Liver (carcinoma)	Huh7	Hanging drop
Sauer SJ, 2017 [132]	Breast (carcinoma), Breast (adenocarcinoma)	SUM149, SUM190, T47D, MCF-7	Forced-floating (ULA)
Shaheen S, 2016 [133]	Colon (carcinoma)	HCT-116	Forced-floating (not defined)
Shen K, 2014 [134]	Breast (adenocarcinoma)	MDA-MB-231	Scaffold-based methods
Sheth DB, 2019 [135]	Breast (adenocarcinoma)	MCF-7	Recent advances
Shortt RL, 2023 [136]	Colon (carcinoma)	HCT-116	Scaffold-based methods
Singh A, 2020 [137]	Breast (adenocarcinoma)	MCF-7	Scaffold-based methods
Suhito IR, 2021 [138]	Bone (neuroblastoma), Brain (glioblastoma)	SH-SY5Y, U-87	Recent advances
Tanenbaum LM, 2017 [139]	Ovary (adenocarcinoma)	UCI101, A2780	Forced-floating (not defined)
Tang S, 2017 [140]	Colon (adenocarcinoma), Ovary (adenocarcinoma)	HT-29, SKOV-3	Hanging drop
Taubenberger AV, 2019 [141]	Breast (adenocarcinoma)	MCF-7	Scaffold-based methods
Terrones M, 2024 [142]	Lung (adenocarcinoma)	HCC78	Forced-floating (ULA)
Tevis KM, 2017 [143]	Breast (adenocarcinoma)	MDA-MB-231	Scaffold-based methods
To HTN, 2022 [144]	Stomach (carcinoma)	SNU-216, SNU-484, SNU-601, SNU-638, SNU-668, and SNU-719	Forced-floating (ULA)
Torisawa YS, 2007 [145]	Breast (adenocarcinoma), Liver (carcinoma)	MCF-7, HepG2	Recent advances
Uematsu N, 2018 [146]	Breast (adenocarcinoma)	MCF-7	Recent advances
Varan G, 2021 [147]	Lung (carcinoma), Liver (carcinoma)	A549, HepG2	Forced-floating (poly-HEMA)
Vinci M, 2012 [148]	Brain (glioblastoma), Oral (carcinoma), Breast (adenocarcinoma)	U87, KNS42, LICR-LON-HN4, MDA-MB-231	Forced-floating (ULA), Agarose plates
Wan X, 2016 [149]	Colon (adenocarcinoma), Ovary (adenocarcinoma)	DLD-1, NCI/ADR	Scaffold-based methods
Wang Y, 2014 [150]	Cervix (adenocarcinoma)	HeLa	Scaffold-based methods
Ware MJ, 2016 [151]	Pancreas (carcinoma)	PANC-1, AsPc-1, BxPC-3, Capan-1, MIA PaCa-2 cells	Hanging drop
Wen Z, 2013 [152]	Pancreas (carcinoma)	MIAPaCa-2, PANC-1	Scaffold-based methods
Wenzel C, 2014 [153]	Breast (carcinoma)	T47D	Forced-floating (liquid overlay)
Weydert Z, 2020 [154]	Ovary (adenocarcinoma)	HEY, SKOV-3	Hanging drop
Wu G, 2019 [155]	Liver (carcinoma)	HepG2, Huh7	Scaffold-based methods
Wu KW, 2020 [156]	Bladder (carcinoma), Lung (carcinoma), Liver (carcinoma)	T24, A549, Huh-7	Recent advances
Xia H, 2020 [157]	Brain (glioblastoma)	LN229, U87	Scaffold-based methods
Xiong Q, 2023 [158]	Bladder	Patient-derived	Forced-floating (ULA)
Yamawaki K, 2021 [159]	Ovary	Patient-derived	Forced-floating (ULA)
Yoshida T, 2019 [160]	Bladder	Patient-derived	Scaffold-based methods
Yu L, 2015 [161]	Breast (adenocarcinoma)	MCF-7	Recent advances
Yu Q, 2021 [162]	Breast (adenocarcinoma)	MDA-MB-436, MDB-MB-231	Scaffold-based methods
Zhang JZ, 2012 [163]	Colon (adenocarcinoma), Ovary (teratocarcinoma)	DLD-1, PA-1 ovarian cancer cells	Forced-floating (liquid overlay)
Zhang JZ, 2012 [164]	Colon (adenocarcinoma)	DLD-1	Forced-floating (liquid overlay)
Zhang X, 2005 [165]	Breast (adenocarcinoma)	MCF-7	Recent advances
Zuchowska A, 2017 [166]	Liver (carcinoma)	HepG2	Recent advances

**Table 3 ijms-26-06478-t003:** Summary of the five most common spheroid sources.

Spheroid Formation Method	Number of Studies	Percentage of Total Number of Studies
Breast	46	24.6%
Colon	23	12.3%
Lung	20	10.7%
Ovary	18	9.6%
Brain	18	9.6%

**Table 4 ijms-26-06478-t004:** Summary of the five most common spheroid seeding methods.

Spheroid Formation Method	Number of Studies	Percentage of Total Number of Studies
Forced-floating	70	31.8%
Scaffold-based	41	18.6%
Recent advances	21	9.5%
Hanging drop	14	6.4%
Suspension culture	4	1.8%

**Table 5 ijms-26-06478-t005:** Summary of advantages and limitations of different seeding methods.

Method	Advantages	Limitations
Forced-floating method	Simple and scalable Produces uniform spheroids Compatible with automation Cost-effective	Spheroid integrity varies with cell density Limited nutrient and oxygen diffusion for large spheroids
Scaffold-based	Mimics ECM Supports long-term cultures Allows co-cultures Highly physiological	Variability in size and shape High cost of scaffolds Complexity in interpreting scaffold-induced effects
Hanging drop method	Low cost Produces uniform spheroids Minimal mechanical damage Control over spheroid size	Labor-intensive Limited scalability Potential damage during transfer Limited nutrient exchange
Suspension culture	Versatile Simple and cost-effective Suitable for high-throughput applications Closer mimicry of in vivo conditions	Nutrient and oxygen limitations in large spheroids Consistency challenges in terms of spheroid size
Microencapsulation	Provides controlled microenvironment Supports co-cultures Protects from shear stress	Nutrient diffusion limitations Challenges in retrieving spheroids
Bioprinting	Precise spheroid placement High control over architecture Effective for high-throughput applications	Requires optimization of bioinks Expensive equipment
Nanoparticle-assisted techniques	Enable precise aggregation Integrate with imaging/therapeutics Reproducible	Potential cytotoxicity of nanoparticles Scalability concerns
Microfluidics and lab-on-a-chip	Controlled environment Mimics in vivo gradients Real-time monitoring Supports co-cultures	High cost Requires technical expertise Limited scalability
Magnetic levitation	High control over size Supports co-cultures Possible to adapt to high-throughput applications Rapid and reproducible	Potential toxicity of nanoparticles Inconsistencies in particle uptake Equipment costs

## Data Availability

The original contributions presented in this study are included in the article. Further inquiries can be directed to the corresponding author.

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
