# Peer review of "Unlocking the Potential of Spheroids in Personalized Medicine: A Systematic Review of Seeding Methodologies"

_ijms, 2025, doi:10.3390/ijms26136478_

Round 1

Reviewer 1 Report

Comments and Suggestions for Authors

The authors evaluate the methods for spheroid formation and cellular sources and highlight diverse applications and preferences in the field. Overall, this review paper must be re-written before consideration for publication.

  1. The introduction should be integrated to discuss the background and the content of this work.
  2. The previous reviewer papers about this field should be discussed in Introduction and the differences with this work should be mentioned.
  3. Part 2 is “Materials and Methods”, and Part 3 is “Results”. This is not the format of review paper.
  4. The conclusions should be shortened and refined to highlight the importance and future of this topic.

Author Response

I would like to sincerely thank you for your thorough and constructive review of my manuscript. I truly appreciate each comment and suggestion you provided, as they have significantly contributed to improving the quality and clarity of the work. I have carefully addressed every point raised and have revised the manuscript accordingly, striving to incorporate your feedback to the best of my ability.

Opinion 1: The introduction should be integrated to discuss the background and the content of this work.

Response: I highly appreciate Introduction was re-written to discuss the background and the content of this work.

Opinion 2: The previous reviewer papers about this field should be discussed in Introduction and the differences with this work should be mentioned.

Response: References to other systematics reviews have been added

Opinion 3: Part 2 is “Materials and Methods”, and Part 3 is “Results”. This is not the format of review paper.

Response: Part “Materials and Methods” was re-named “Methods”. Section “Results” summarize results from systematic review and in my opinion name of this section should remain “Results”. However, I am open to any further suggestions.

Opinion 4: The conclusions should be shortened and refined to highlight the importance and future of this topic.

Response: Conclusions were re-written to highlight the importance and future of this topic and shortened. “Future perspectives” section was added.

Reviewer 2 Report

Comments and Suggestions for Authors

In the manuscript entitled “Unlocking the Potential of Spheroids in Personalized Medicine: Systematic Review of Seeding Methodologies,” the authors present important findings compiled from various research articles. Despite the manuscript being well written, there are several points that should be addressed before recommending acceptance for publication:

  1. I recommend revising the data presented in the tables to include specific references for the information cited (specifically in Table 1, Table 3, Table 4, and Table 5).

  1. I suggest enhancing the figure descriptions to provide readers with a clearer understanding of the images presented.

  1. Some relevant information described in Section 2, Materials and Methods, is not mentioned in the abstract. It is important to include this information so that readers can gain perspective on the publication dates of the analyzed documents. Therefore, I recommend rewriting the abstract to reflect this detail.

  1. In the Results section, very interesting information regarding spheroids is described. I suggest including some representative images from the cited research articles (with appropriate permissions) to visually illustrate the spheroid structures. This would significantly enhance the usability and impact of the review.

  1. I recommend adding a Perspectives section, where the authors propose future research directions based on the bibliography reviewed.

Author Response

I would like to sincerely thank you for your thorough and constructive review of my manuscript. I truly appreciate each comment and suggestion you provided, as they have significantly contributed to improving the quality and clarity of the work. I have carefully addressed every point raised and have revised the manuscript accordingly, striving to incorporate your feedback to the best of my ability. Below are presented my answers:

Opinion 1: I recommend revising the data presented in the tables to include specific references for the information cited (specifically in Table 1, Table 3, Table 4, and Table 5).

Response: Specific referenced were included in the Table 1, 3 and 4. Table 5 summarize advantages and limitations of spheroid seeding methods that are described in solid text earlier.

Opinion 2: I suggest enhancing the figure descriptions to provide readers with a clearer understanding of the images presented.

Response: Figure descriptions in solid text was enhanced to provide clearer understanding.

Opinion 3: Some relevant information described in Section 2, Materials and Methods, is not mentioned in the abstract. It is important to include this information so that readers can gain perspective on the publication dates of the analyzed documents. Therefore, I recommend rewriting the abstract to reflect this detail.

Response: Abstract was re-written to reflect detail from Materials and Methods section.

Opinion 4: In the Results section, very interesting information regarding spheroids is described. I suggest including some representative images from the cited research articles (with appropriate permissions) to visually illustrate the spheroid structures. This would significantly enhance the usability and impact of the review.

Response: I am not sure what specific information do you found interesting. Can you indicate what information from result do you mean? Illustration of spheroid seeding method is presented in figure 5.

Opinion 5: I recommend adding a Perspectives section, where the authors propose future research directions based on the bibliography reviewed.

I strongly agree with your suggestion. Conclusion section was re-written and Perspective section was added.

Reviewer 3 Report

Comments and Suggestions for Authors

I would like to express my sincere gratitude to the editor for inviting me to review this article. The author focuses on the application of 3D tumor spheroid models in personalized medicine, and the article holds significant scientific and clinical significance. The systematic review methodology is rigorous, the database screening and inclusion criteria are clear, and the data in the results section are substantial. In my opinion, it is suitable for publication. However, several errors need to be further corrected before publication.

1. Table 3 does not conform to the content in the main text.
2. In Table 3, Magnetic levitation has not been reported by any article. What is the significance of the author including it here?
3. Has the author considered not piling up several references at the very end of each paragraph? It would be better to make each reference more relevant.
4. The description in the part of "Recent Advances in Spheroid Seeding Methods, Line 406-429" is too general. It is recommended to supplement specific cases or technical details.
5. In the discussion section, it is advisable to explore more in-depth the limitations of different spheroid formation methods.
6. The format of the references needs to be uniformly checked (for example, the abbreviations of some journal names are inconsistent).
7. For some abbreviations (such as ULA, PLGA), their full names should be marked when they first appear. 

Author Response

I would like to sincerely thank you for your thorough and constructive review of my manuscript. I truly appreciate each comment and suggestion you provided, as they have significantly contributed to improving the quality and clarity of the work. I have carefully addressed every point raised and have revised the manuscript accordingly, striving to incorporate your feedback to the best of my ability.

Opinion 1: Table 3 does not conform to the content in the main text.

Response: I made a mistake when transferring a table from the target file. Table 3 has been checked and corrected, after correction it is consistent with the main content.

Opinion 2: In Table 3, Magnetic levitation has not been reported by any article. What is the significance of the author including it here?

Response: I agree with your suggestion, including information about magnetic levitation in the table was unnecessary. This table row was removed.

Opinion 3. Has the author considered not piling up several references at the very end of each paragraph? It would be better to make each reference more relevant.

Response: I agree with the suggestion. Where deemed appropriate, references have been highlighted more and added in places as confirmation of statements.

Opinion 4. The description in the part of "Recent Advances in Spheroid Seeding Methods, Line 406-429" is too general. It is recommended to supplement specific cases or technical details.

Response: Mentioned fragment was rewritten and some specific cases an technical details were added.

Opinion 5. In the discussion section, it is advisable to explore more in-depth the limitations of different spheroid formation methods.

Response: Conclusion section was re-written, summary of seeding methods is presented in Table 5.

Opinion 6. The format of the references needs to be uniformly checked (for example, the abbreviations of some journal names are inconsistent).

Response: References were checked and uniformed.

Opinion 7. For some abbreviations (such as ULA, PLGA), their full names should be marked when they first appear. 

Response: Full names were introduced for abbreviations (when they first appeared).

Reviewer 4 Report

Comments and Suggestions for Authors

This systematic review explores the use of 3D tumor spheroid models in personalized medicine, focusing on spheroid formation methods (e.g., forced-floating, scaffold-based, hanging drop) and their applications in cancers like breast, colon, and lung. Forced-floating and scaffold-based methods are most used, but challenges in reproducibility and standardization are highlighted. The review emphasizes spheroids’ potential to mimic tumor microenvironments for drug screening and personalized therapy, calling for unified methodologies to enhance translational research. It is suggested that it be published after major revision.

  1. Table 3 and Table 4 are same.
  2. The quality of Figure 5 should be increased.
  3. The latest progress of the spheroid seeding method is not introduced in detail enough. Pictures should also be given in this part.
  4. More recent published literatures related to 2D culture should be cited, such as “International Journal of Pharmaceutics: X, 2025, 9, 100334”

Author Response

I would like to sincerely thank you for your thorough and constructive review of my manuscript. I truly appreciate each comment and suggestion you provided, as they have significantly contributed to improving the quality and clarity of the work. I have carefully addressed every point raised and have revised the manuscript accordingly, striving to incorporate your feedback to the best of my ability.

Opinion 1: Table 3 and Table 4 are same.

Response: An error was made when pasting the manuscript into the journal format. The error has been corrected, and the table is discussed in the manuscript body.

Opinion 2: The quality of Figure 5 should be increased

Response: Figure 5 has been added to the manuscript body in higher resolution.

Opinion 3: The latest progress of the spheroid seeding method is not introduced in detail enough. Pictures should also be given in this part.

Response: A more thorough introduction to the "Recent advances" section has been written. The images for this section are shown in Figure 5.

Opinion 4: More recent published literatures related to 2D culture should be cited, such as “International Journal of Pharmaceutics: X, 2025, 9, 100334”

Response: More recent literature about 2D cultures was added.

Round 2

Reviewer 1 Report

Comments and Suggestions for Authors

 Accept in present form

Author Response

Thank you for your thoughtful and constructive comments on my manuscript. I have carefully considered all of your suggestions and greatly appreciate the time and expertise you invested in reviewing my work. Your insights were invaluable in helping me improve the clarity and overall quality of the paper. I am confident that the revised version is significantly strengthened as a result of your feedback.

Reviewer 2 Report

Comments and Suggestions for Authors

The authors have satisfactorily addressed most of the comments; however, there are still some points that need further attention:

Comment 1:

“…

Opinion 2: I suggest enhancing the figure descriptions to provide readers with a clearer understanding of the images presented.

Response: Figure descriptions in solid text were enhanced to provide clearer understanding.

…”

Observation:

I believe the descriptions in the text should be further improved. For example, the description in Figure 1 states “Cellular structure of spheroid.” A more appropriate description might be: “Schematic representation of the cellular structure of a spheroid.”

A careful revision of all figure descriptions by the authors would help readers better understand the figures included in the manuscript.

Comment 2:

“…

Opinion 4: In the Results section, very interesting information regarding spheroids is described. I suggest including some representative images from the cited research articles (with appropriate permissions) to visually illustrate the spheroid structures. This would significantly enhance the usability and impact of the review.

Response: I am not sure what specific information you found interesting. Can you indicate what information from the results you are referring to? An illustration of the spheroid seeding method is presented in Figure 5.

…”

Clarification:

I am referring to the inclusion of relevant experimental results that could serve as a reference for experimental procedures—such as characterizations and representative experimental images obtained through optical or electron microscopy—related to spheroid formation techniques. These examples would significantly improve the quality and utility of the review.

Author Response

Observation:

I believe the descriptions in the text should be further improved. For example, the description in Figure 1 states “Cellular structure of spheroid.” A more appropriate description might be: “Schematic representation of the cellular structure of a spheroid.”

A careful revision of all figure descriptions by the authors would help readers better understand the figures included in the manuscript.

Answer: I agree with your observation. I improved figur captions.

Clarification:

I am referring to the inclusion of relevant experimental results that could serve as a reference for experimental procedures—such as characterizations and representative experimental images obtained through optical or electron microscopy—related to spheroid formation techniques. These examples would significantly improve the quality and utility of the review.

Answer: 

Thank you very much for your clarification — it was extremely helpful. I agree that such examples will improve quality of manuscript.

I am currently in the process of obtaining the necessary permissions from the respective authors and/or publishers to include the images in the final version of the manuscript. I aim to obtain images for every seeding method. Unfortunately, I am unable to include them at this stage, as I do not wish to infringe upon their intellectual property rights.

Reviewer 4 Report

Comments and Suggestions for Authors

This is highly disappointing. The authors failed to adequately address the reviewers' comments, and the manuscript revisions fall far below expectations. Specifically, they neglected to respond to Opinion 4 from the previous review comment and omitted citing the relevant literature as requested.

Author Response

Dear Reviewer,

I sincerely apologize for not addressing the indicated issues as expected. I made an effort to locate the manuscript you referenced; however, I may have overlooked it or encountered difficulties during the search.

I am very open to citing the latest literature and would be happy to read and reference another relevant publication. If possible, could you kindly provide the DOI or full title of the manuscript in question?

Please rest assured that I am fully open to making any necessary corrections to improve the manuscript.

Once again, I apologize for the oversight and truly appreciate your guidance and support.

Round 3

Reviewer 4 Report

Comments and Suggestions for Authors

More recent published literatures related to 2D culture should be cited, such as https://doi.org/10.1016/j.ijpx.2025.100334

Author Response

Dear Reviewer,

Thank you very much for drawing my attention to this important publication, which is highly relevant to the subject matter discussed in my review article. I truly appreciate your thoughtful input and the valuable reference.

Kind regards,

Round 4

Reviewer 4 Report

Comments and Suggestions for Authors

None